# Network Pharmacology Exploration Reveals Anti-Apoptosis as a Common Therapeutic Mechanism for Non-Alcoholic Fatty Liver Disease Treated with Blueberry Leaf Polyphenols

**DOI:** 10.3390/nu13114060

**Published:** 2021-11-13

**Authors:** Cai-Ren Wang, Hong-Wei Chen, Yan Li, Ming-Yue Zhou, Vincent Kam-Wai Wong, Zhi-Hong Jiang, Wei Zhang

**Affiliations:** 1State Key Laboratory of Quality Research in Chinese Medicines, Macau Institute for Applied Research in Medicine and Health, Macau University of Science and Technology, Taipa, Macau 999078, China; cairen1118@gmail.com (C.-R.W.); musthwchen@gmail.com (H.-W.C.); tuantuanyan0304@hotmail.com (Y.L.); yuemingzhou1995@163.com (M.-Y.Z.); bowaiwong@gmail.com (V.K.-W.W.); 2Faculty of Medicine, Macau University of Science and Technology, Taipa, Macau 999078, China

**Keywords:** polyphenols in blueberry leaves, network pharmacology, NAFLD, apoptosis

## Abstract

Non-alcoholic fatty liver disease (NAFLD) is the most common chronic liver disease characterized by excessive fat accumulation in the liver. The aim of this study is to elucidate the multi-target mechanism of polyphenols in blueberry leaves (PBL) on NAFLD by network pharmacology and to validate its results via biological experiments. Twenty constituents in PBL were preliminarily determined by liquid chromatography-tandem mass spectrometry. Subsequently, 141 predicted drug targets and 1226 targets associated with NAFLD were retrieved from public databases, respectively. The herb-compound-target network and the target protein–protein interaction network (PPI) were established through Cytoscape software, and four compounds and 53 corresponding targets were identified. Gene Ontology (GO) enrichment and Kyoto Encyclopedia of Genes and Genomes (KEGG) pathway enrichment were performed to explore the biological processes of the predicted genes. The results of cell experiments demonstrated that PBL could significantly improve the viability of the NAFLD cell model, and the protein expressions of caspase-3 and Bcl-2 were consistent with the expected mechanism of action of PBL. Those results systematically revealed that the multi-target mechanism of PBL against NAFLD was related to the apoptosis pathway, which could bring deeper reflections into the hepatoprotective effect of PBL.

## 1. Introduction

Non-alcoholic fatty liver disease (NAFLD) is regarded as the most common cause of liver disease in many developed countries, and its incidence is also increasing dramatically in developing countries [1,2]. It is characterized by steatosis, inflammation, hepatocellular ballooning, hepatocellular injury, etc., which may progress to non-alcoholic steatohepatitis (NASH), and even induce the development of liver cirrhosis or hepatocellular carcinoma [3,4,5]. Although the pathogenesis of NAFLD is unknown, insulin resistance and genetic factors seem to be pivotal in the pathogenesis, and multiple-hit pathogenesis is the most widely accepted [6,7,8]. Generally, hepatocyte apoptosis is a significant pathological feature of human NAFLD and plays a critical role in the occurrence and development of NAFLD [9,10]. Some apoptosis inhibitors have been considered as potential targets for NAFLD over recent years. Junli Zhang et al. [11] reported that baicalin (BA) significantly inhibited hepatocyte apoptosis in methionine and choline-deficient (MCD) diet-induced mice by reducing apoptotic cells and the protein level of caspases-3. Resveratrol (RSV) could significantly decrease the content of lipid peroxidation, inflammatory cytokines and apoptotic cells, thus significantly improved liver injury [12]. Therefore, strategies aimed at reducing apoptosis may lead to better treatment for NAFLD.

Nutritional therapy and phytotherapy have been used as treatment for medical conditions for thousands of years [13]. Blueberries are common fruits that are rich in flavonoids, especially polyphenolic phytochemicals with anti-oxidative, anti-inflammatory and immune regulatory activities [14,15]. Blueberry leaves are byproducts of the blueberry industry. Several studies demonstrated that the blueberry leaf extracts have a strong antibacterial, antioxidant and antiviral activity [16,17,18]. Those extracts can be used as dietary supplements for the prevention of ailments associated with metabolic syndrome [19]. Blueberry leaves are also rich in edible value. As a safe and convenient way of eating, blueberry leaves, tea is composed of water, blueberry leaves and fructose syrup as raw materials, without any additives [20]. In addition, green, oolong, and black tea can be extracted from blueberry leaves [18]. We also found that polyphenols in blueberry leaves (PBL) could improve mitochondrial dysfunction and oxidative defense through activation of AMPK/PGC-1α/SIRT3 signaling axis, thereby reduce liver steatosis, oxidative stress as well as inflammation, and eventually alleviate NAFLD [21]. However, the bioactive compounds and underlying pharmacological mechanisms still remain unclear due to its complex composition. Network pharmacology could exhibit complex data interactions through visual node interactions, especially in analyzing the relationship between drugs and diseases to reveal the synergy of multi-molecule drugs. Therefore, this method is often used to analyze the “drug-component-target-disease” interaction network [22]. In this study, the putative active ingredients and underlying mechanism of PBL on NAFLD were comprehensively investigated using network pharmacology and verified with the aid of biological experiments. The results obtained from this study could be recommended as a supplement for the phytotherapy of PBL.

## 2. Materials and Methods

### 2.1. Materials and Reagents

Human hepatocellular cancer cell line (HepG2) was purchased from American Type Culture Col-lection (ATCC; Rockville, MD, USA). Blueberry leaves were bought from Nanjing Jinrui Blueberry Professional Cooperative. Dulbecco’s modified Eagle’s medium (DMEM), 0.25% trypsin-ethylene diamine tetraacetic acid (EDTA) solution, fetal bovine serum (FBS), and penicillin-streptomycin solution were acquired from GIBCO (Grand Island, NY, USA). 3-[(4,5)-Dimethylthiazol-2-yl]-2,5-diphenyl tetra-zolium bromide (MTT), bovine serum albumin (BSA), phosphate buffer saline (PBS), sodium palmitate and Oil Red O and were purchased from Sigma Aldrich Chemical Co. (St. Louis, MO, USA). Deionized water was obtained from a Milli-Q Gradient Water System (Millipore Corp., Bedford, MA, USA). In addition, The Muse TM Annexin V and Dead Cell Kit was bought from Merck Limited (Darmstadt, Germany). Bradford reagent was obtained from Bio-Rad Laboratory (Hercules, CA, USA). RIPA lysis buffer, antibodies against caspase-3 and Bcl-2 were bought from Cell Signaling Technology Inc. (Boston, MA, USA), β-actin and secondary antibody were provided by Abcam Inc. (Cambridge, UK).

### 2.2. Screening for Active Components and Target Genes

Information on the main compounds in PBL was acquired according to the previous research [21]. The targets of PBL were mainly obtained from the Traditional Chinese Medicine System Pharmacology Database (TCMSP, https://old.tcmsp-e.com/tcmsp.php, accessed on 11 February 2021) and STITCH (http://stitch.embl.de/, accessed on 11 February 2021) [23,24]. The TCMSP database contains the affinities between absorption, distribution, metabolism, and excretion (ADME) parameters of each component. OB refers to “the rate and extent to which the active ingredient or active moiety is absorbed from a drug product and becomes available at the site of action” [25]. DL is a qualitative concept for drug design, which has the ADME properties of components and known drugs [26]. The effective components of blueberry leaves were selected based on the demands of both OB ≥ 30% and DL ≥ 0.18 [23]. Canonical gene names and UniProt IDs of all the targets were standardized according to the UniProtKB (https://www.uniprot.org/, accessed on 11 February 2021) database [27].

### 2.3. Screening of Disease Targets

The targets were obtained from the following databases: (1) Online Mendelian Inheritance in Man (OMIM) (http://omim.org/, updated 1 October 2020, accessed on 11 February 2021), an online catalog mainly designed for human genes and genetic diseases [28]. (2) The Human Gene Database (GeneCards) (https://www.genecards.org, updated on 11 March 2020, accessed on 11 February 2021), which provides comprehensive, user-friendly information on all annotated and predicted human genes [29]. (3) Drugbank database (https://www.drugbank.ca, last updated by 20 December 2020, accessed on 11 February 2021), which offers target genes from FDA-approved drugs [30]. All genes were retrieved from the database using the keywords “NAFLD” and “non-alcoholic fatty liver disease”. After merging the targets in the three diseases database, the duplicates were then removed.

### 2.4. Searching for Common Targets and Key Targets of Polyphenols in Blueberry Leaves (PBL) and Non-Alcoholic Fatty Liver Disease (NAFLD)

Firstly, the common targets of drug and disease were found through the Venn Diagrams Draw (http://bioinformatics.psb.ugent.be/webtools/Venn/, accessed on 12 February 2021). Then, the common targets were entered into STRING (http://string-db.org/, version 11.5) for protein–protein interaction (PPI) analysis. The STRING database was designed to integrate the associations between all known and predicted proteins. The data used to predict PPIs come from multiple sources. Then, the analysis mode was set to “multiple proteins”, and the species was limited to “*Homo sapiens*”. In this study, the interactions with probabilistic association confidence score ≥7 were selected [31,32]. The protein interaction relationship was obtained and exported as a tab-separated value (TSV) format. The PPI analysis network was depicted through Cytoscape software, the size of node could indicate the degree size and the composite score is expressed by the thickness of edge. To further explore the molecular mechanism of PBL in the treatment of NAFLD, the online functional annotation and enrichment tool DAVID (https://david.ncifcrf.gov/, accessed on 12 February 2021) was used for Gene Ontology (GO) analysis and Kyoto Encyclopedia of Genes and Genomes (KEGG) pathway enrichment of protein targets, the organism was set to “*Homo sapiens*” and the results with *p* < 0.05 were considered statistically significant.

### 2.5. Effect of PBL on NAFLD-Modeled Cells

#### 2.5.1. Cell Culture

HepG2 were cultured in DMEM supplemented with 10% FBS, 100 µg/mL streptomycin and 100 U/mL penicillin, in the presence of 5% CO_2_—5% air at 37 °C humidified incubator. After culturing for 24 h, cells were exposed to palmitic acid (PA) in order to induce the NAFLD-modeled cells. After that, the cells were treated with PBL at the demanded concentrations for 48 h. The PA dissolved in 10% BSA at 70 °C to make a final stock of 10 mM. The concentration and time point for NAFLD-modeled cell treatment were based on MTT cytotoxicity analysis.

#### 2.5.2. MTT Analysis

MTT analysis method was adopted to analyzed cell viability. Briefly, HepG2 cells were add to 96 wells plate (3500 cells/well). After 24 h of incubation, HepG2 cells were exposed to a series concentration of PA (0.1, 0.2, 0.3, 0.5, 0.8, 1 mM) for 48 h. To detect the effect of PBL on cell viability, HepG2 cells were exposed to a range of PBL concentrations (0.78, 1.59, 3.18, 6.25, 12.5, 25 µg/mL) for 48 h. Moreover, we focused on the impact of PBL on the survival rate of HepG2 cells induced by PA, cells were cultured with 300 µM PA and PBL from 0.78 to 25 µg/mL at 48 h. To add 10 µL MTT solution to each well and continue incubation for 4 h. The medium was removed, and 200 µL of DMSO was added to each well, then the absorbance was measured at 570 nm using Multi-Mode Microplate Detection Platform (Molecular Devices, Sunnyvale, CA, USA).

#### 2.5.3. Oil Red O Staining in HepG2 Cells

HepG2 cells were seeded in a 6-well plate (10,000 per well) for 24 h. The cells were treated with 300 µM PA and PBL (6.25, 12.5, 25 µg/mL) for 48 h. All cells were fixed with 4% paraformaldehyde for 15 min at room temperature. After that, cells were washed twice with PBS, stained with Oil Red O (50 g/mL) working solution for 15 min and examined under an optical microscope (×10 magnifications).

#### 2.5.4. Cell Apoptosis Analysis

Cells were treated with PA and different PBL concentrations (6.25, 12.5, 25 µg/mL) for 48 h, after trypsinization and washed twice with 4 °C PBS. The treated cells were resuspended in 1% FBS and stained with Muse TM Annexin V and Dead Cell reagent (Muse TM cell analyzer, Merck Millipore, Darmstadt, Germany), then subjected to flow cytometry to quantify the rate of apoptosis

#### 2.5.5. Western Blot Analysis

HepG2 cells were treated with PA and different concentrations of the PBL (6.25, 12.5, 25 µg/mL). After 48 h, the cellular total proteins in each group were prepared in RIPA lysis buffer. Protein concentration was determined by Bradford reagent. After that, cell lysates were mixed with 5 × SDS-loading buffer (4:1, *v*/*v*) and heated at 100 °C with locked capping for 5 min. The cell lysates were subjected to 10% sodium dodecyl sulphate–polyacrylamide gel electrophoresis (SDS-PAGE). After electrophoresis, the protein from SDS-PAGE was transferred to PVDF membranes, the membranes were sealed with 5% non-fat dry milk (*w*/*v*) for 1 h, and then it was incubated with antibodies against caspase-3, Bcl-2 and β-actin at 4 °C. The aforementioned PVDF membranes were incubated with second antibodies for 1 h. The bands were visualized and quantitated using the Image J 1.46r software (National Institutes of Health, Bethesda, MD, USA)

### 2.6. Statistical Method

All the data were analyzed using the IBM SPSS Statistics 25.0 Software and GraphPad Prism 9.0 Software. All results were expressed in the study represent as mean ± SEM from three independent replicate experiments. Values of *p* < 0.05 were considered to be statistically significant.

## 3. Results

### 3.1. Screening of Effective Components of PBL

Preparation and quantification of PBL were performed according to previously published methods [21]. A total of 20 compounds were identified and the 8 polyphenols were simultaneously quantified in blueberry leaf extracts. The contents of chlorogenic acid, (+)-catechin, (−)-epicatechin, rutin, isoquercitrin, cyanidin-3-O-glucoside, iridin and quercetin in PBL (10 µg/mL) were 875 ± 22.6, 84.1 ± 10.9, 96.1 ± 6.9, 287.5 ± 18.2, 410.3 ± 14.3, 100.6 ± 5.4, 17.1 ± 2.2, 217.3 ± 12.2 nM, respectively. OB ≥ 30% and DL ≥ 0.18% were used as the screening criteria to obtain four compounds, including procyanidin B1, catechin, quercetin and kaempferol.

### 3.2. Target Recognition Results

Genes related to NAFLD were searched by using GeneCards and Drugbank database, and a total of 1226 targets with scores ≥50 was screened. There were 141 drug targets retrieved from the TCMSP database. After that, the intersection of drug targets and disease targets was conducted. Then, 53 key target genes for PBL in the treatment of NAFLD were obtained based on the Venn diagram. The target recognition was carried out using the Venny 2.1.0 online system, and the results were shown in Figure 1a.

### 3.3. Protein–Protein Interaction (PPI) Network Construction and Module Mining

The 53 targets of PBL in the treatment of NAFLD were imported into the STRING 11.0 database in order to obtain PPI (Figure 1b). The PPI results were exported as simple textual data format (.tsv), and the TSV file was imported into Cytoscape 3.8.2 to acquire the network diagram of target interaction. The average value of all points (Degree, Betweenness centrality and Closeness centrality) are the calculation results and three topology parameters were obtained: 24.05, 0.04 and 0.65 respectively. In addition, AKT1, IL6, VEGFA, TNF, CAT, CASP3, JUN, ESR1, PTGS2 and PPARG represented the crucial targets of PBL based on the degree and betweenness centrality. Herb-compound-target network of PBL was composed of 59 nodes and 127 edges in total (Figure 2). This section may be divided by subheadings. It should provide a concise and precise description of the experimental results, their interpretation, as well as the experimental conclusions that can be drawn.

### 3.4. Gene Ontology (GO) Function Enrichment and Kyoto Encyclopedia of Genes and Genomes (KEGG) Enrichment Analysis

We imported 53 targets into the DAVID 6.8 database, the GO function and KEGG signal pathway were screened according to *p* < 0.05 and FDR < 0.05. The characteristics of PBL-related targets were investigated by GO enrichment and KEGG pathway enrichment analysis. The GO function represents three aspects of biology, molecular functions (MF), biological processes (BP) and cellular components (CC). GO enrichment analysis showed that target genes were mostly related to the BPs of positive regulation of nitric oxide biosynthetic process, negative regulation of apoptotic process and response to hypoxia, etc. The enriched MF ontologies were dominated by protein binding, enzyme binding, protein homodimerization activity, and so on. The cytoplasm was the biggest proportion in CC analysis base on adjusted percent of genes (Figure 3). Finally, KEGG analysis revealed that the active ingredients of PBL could affect multi-pathways, including chemical carcinogenesis, TNF signaling pathway, drug metabolism—cytochrome P450 and pathways in cancer, etc. (Figure 4). The cytochrome P450 (CYP) was additional variants in regulatory genes or in NADPH, downregulation of CYP3A4 protein and activity in NAFLD [33,34]. Enhancement of TNF–α signalling may be critical in the pathogenesis of hepatic steatosis and fibrosis, blockade of TNFR1/TNFR2 signalling is a promising therapeutic target for NAFLD [35].

### 3.5. Effect of PBL on the Viability and Lipid Accumulation of NAFLD-Modeled Cells

MTT analysis showed that cell viability presented a decreasing tendency with the increasing concentrations of PA from 100 µM to 1000 µM for 48 h. The maximum inhibition (36–40%) was observed with the highest concentration of PA (Figure 5a). Moreover, Oil Red O staining was performed to investigate the intracellular lipid accumulation of HepG2 cells. Finally, PA 300 µM has been selected for the following study because of its moderate toxicity and lipid accumulation in HepG2 cells. On the other hand, PBL at the concentration of 0.78 to 25 µg/mL showed no cytotoxicity to HepG2 cells (Figure 5b). In addition, the impact of PBL on the viability of NAFLD-modeled cells was also investigated. Figure 5c showed cells treated with PBL for 48 h presented increasing viability compared with NAFLD-modeled cells. As shown in Figure 5d, PA treatment (300 µM for 48 h) could increase the lipid accumulation in HepG2 cells by which the pretreatment with PBL (12.5 or 25 µg/mL) could attenuate this accumulation in a dose-dependent manner.

### 3.6. Cell Apoptosis Assay

The effect of PBL on cell apoptosis was also examined using flow cytometry. The number of apoptotic cells increased sharply with the values of 17.98% after being induced with PA at 300 µM for 48 h. By contrast, PBL treatment with the concentration of 6.25 and 25 µg/mL could significantly decrease the early apoptosis population versus NAFLD-modeled cells in a dose-dependent manner (Figure 6a). These results indicated that PBL treatment could suppress apoptosis induced by NAFLD.

### 3.7. Effect of PBL on the Protein Expression in Palmitic Acid (PA)-Induced Cells

According to the apoptosis results, PBL could effectively alleviate the early apoptosis of NAFLD-modeled cells and increase the proportion of living cells. Therefore, the protein expression of Bcl-2 and caspase-3 were measured by Western blot analysis to clarify the PBL protective effect on NAFLD-modeled cells was related to the apoptosis pathway. There was significant reduction in Bcl-2 protein expression in PA treated cells while caspase-3 protein expression was up-regulated. At the same time, PBL (25 µM) could significantly down-regulate caspase-3 protein expression and slightly up-regulated anti-apoptotic protein Bcl-2 (Figure 6b,c).

## 4. Discussion

Network pharmacology is considered to be a novel and suitable tool to discover bioactive ingredients and action mechanisms of traditional Chinese medicine [36]. In this study, the effective components and therapeutic targets of PBL were screened by network pharmacology. Based on the hub nodes-compound network, four active compounds in PBL, 141 compound-related targets, and 1226 NAFLD-related targets were identified from public databases. Among these, 53 targets shared compound-related and NAFLD-related targets, implicating PBL was likely to plays a protective role in NAFLD. By combining the network contribution values with the visual network map, the procyanidin B1, catechin, quercetin and kaempferol were closely related to these targets, and quercetin has the highest correlation. Accumulating studies have shown that quercetin is a therapeutic approach for NAFLD via its anti-inflammatory and antioxidant [37]. Previous studies showed that quercetin could reverse gut microbiota imbalance and related endotoxemia-mediated toll-like receptor 4 (TLR-4) pathway induction. The anti-inflammatory activity of quercetin might involve in its blockage of lipid metabolism gene expression deregulation [38]. Meanwhile, Kanda et al. [39] reported that quercetin could decrease hepatic intracellular triglycerides (TG) content, promote hepatic very low-density lipoprotein (VLDL) assembly and lipophagy by activating the IRE1a/XBP1s pathway to alleviate NAFLD. According to the literature reports, the other three compounds also have plentiful biological activities. Procyanidin B1, kaempferol and catechin displayed anti-inflammatory, anti-oxidation and anticancer effect [40,41,42]. These studies suggest that PBL has strong support in NAFLD treatment via network pharmacology.

Apoptotic hepatocytes could stimulate the progression of immune cells and hepatic stellate cells to liver fibrosis through the production of inflammasomes and cytokines [43]. Thus hepatocyte apoptosis plays a critical role in the occurrence and development of NAFLD. Hepatocyte apoptosis is significantly increased in patients with NASH and correlates with disease severity [44]. In this study, KEGG pathway analysis indicated that the action mechanism of PBL in the treatment of NAFLD might have a close relationship with the pathways, including chemical carcinogenesis, TNF signaling pathway and drug metabolism—cytochrome P450. In particular, apoptosis pathway might be highly associated with the ingredients of PBL and NAFLD, which are consistent with previous reports.

A variety of intracellular signal transduction pathways have been demonstrated to set off hepatocyte apoptosis in NAFLD, and the activation of caspases and Bcl-2 family proteins also participated in the apoptosis induced by NAFLD [43]. Biochemical processes in apoptosis are accomplished through either the intrinsic or extrinsic pathways, Bcl-2 family proteins can active their pro-apoptotic and anti-apoptotic proteins, which may be implicated in the regulation of apoptosis [45]. Our findings indicated that PBL could slightly up-regulated the expression of Bcl-2 which is an anti-apoptotic protein. Furthermore, many works in the literature have reported that the caspase family could activate cell destruction mechanisms related to apoptosis signaling pathways. Among them, caspase-3 is a secret factor in apoptosis execution, which could be cleaved and activated by the downstream substrate procaspase-3 [46,47]. Certainly, PBL administration significantly down-regulated the expression of caspase-3 protein. Therefore, the regulatory effect of PBL on NAFLD may be related to cell apoptosis.

In conclusion, PBL has a potential therapeutic effect on NAFLD according to the network pharmacological analysis. Although PBL could possibly act on NAFLD via inhibiting the apoptotic pathway, there are still some limitations of in vitro study since the bioavailability of some polyphenols are moderate. It is also important to investigate their pharmacological mechanisms via in vivo study. Hence, we will focus on the relevant in vivo exploration and provide deeper insight into the hepatoprotective of PBL in further study.

## Figures and Tables

**Figure 1 nutrients-13-04060-f001:**
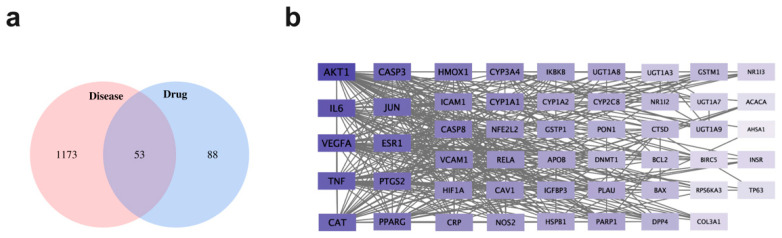
The targets related to polyphenols in blueberry leaves (PBL) in the treatment of non-alcoholic fatty liver disease (NAFLD). (**a**) Distribution of PBL potential targets and NAFLD targets. (**b**) Protein–protein interaction (PPI) networks of PBL for the treatment of NAFLD. The importance of targets depends on the shade of purple and the grey line represents the interaction relationship.

**Figure 2 nutrients-13-04060-f002:**
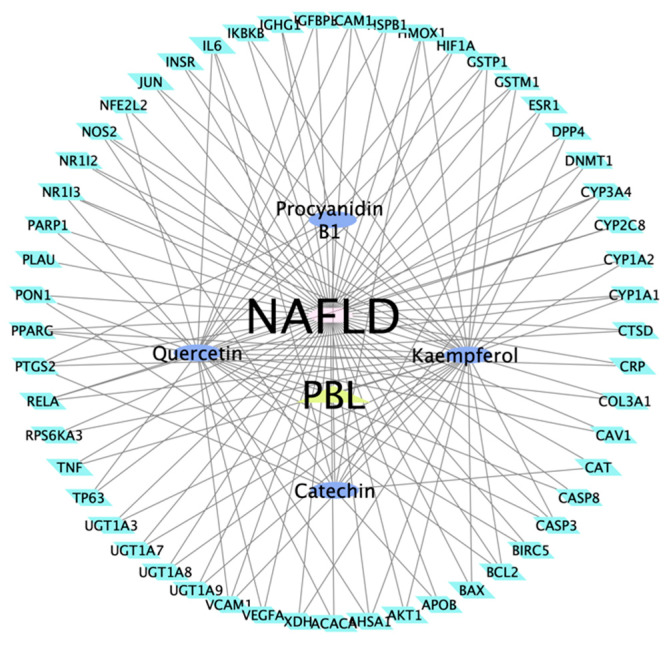
Herb-compound-target network of PBL.

**Figure 3 nutrients-13-04060-f003:**
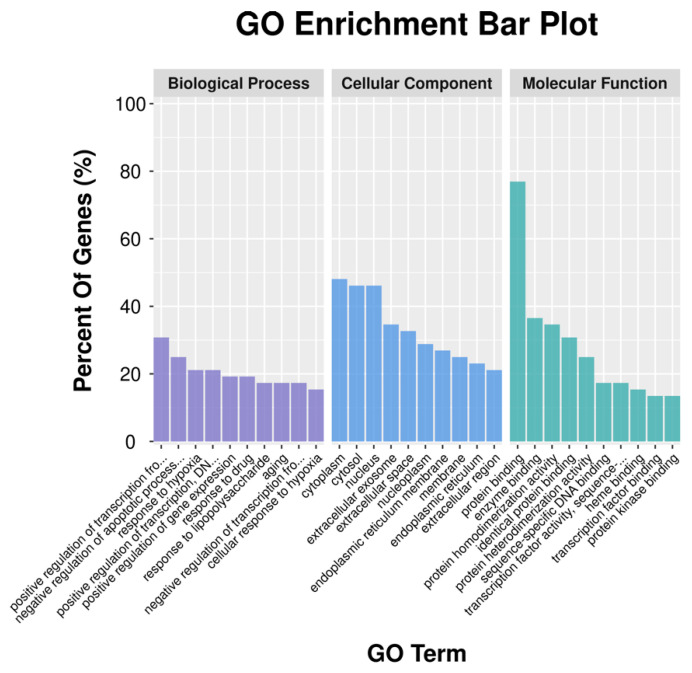
Gene Ontology (GO) enrichment analysis for potential targets of PBL.

**Figure 4 nutrients-13-04060-f004:**
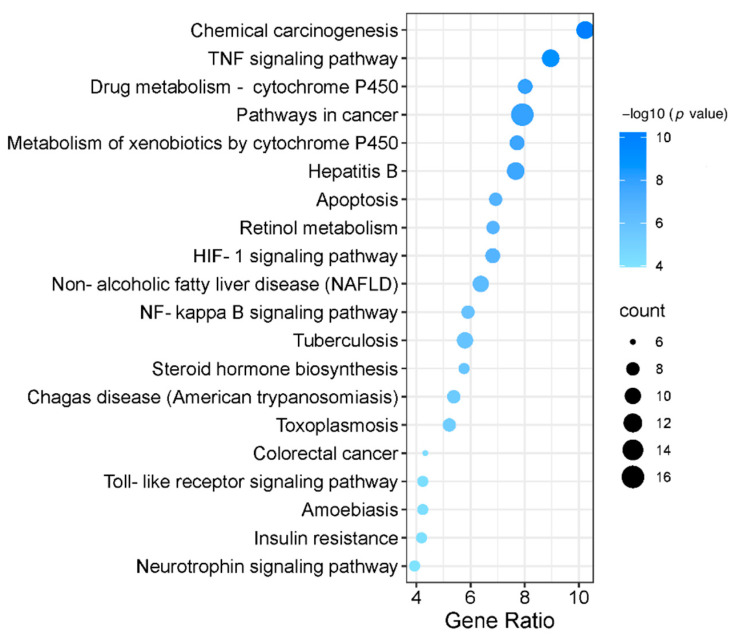
Kyoto Encyclopedia of Genes and Genomes (KEGG) pathway enrichment of PBL against NAFLD. Top 20 pathways enriched based on target genes (the abscissa is the gene ratio, the ordinate is pathway name, the size of the dot indicates the number of target genes, and the color represents the *p* value).

**Figure 5 nutrients-13-04060-f005:**
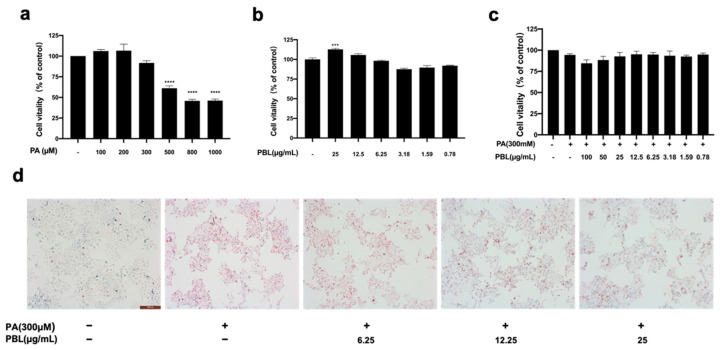
The effect of PBL on the viability of NAFLD-modeled cells. (**a**) Effects of palmitic acid (PA) on cell viability. HepG2 cells were incubated with different concentrations of PA for 48 h. (**b**) The result of PBL (0.78–100 µg/mL) on cell viability after 48 h of exposure. (**c**) Effect of PBL on PA-induced cell viability reduction. HepG2 cells were treated with PBL (0.78–100 µg/mL) and induced by PA (300 µM) for 48 h. (**d**) HepG2 cells evaluated by Oil Red O staining. The data represent as mean ± SEM for *n* = 3 studies. *** *p* < 0.001, **** *p* < 0.0001 compared with the control group.

**Figure 6 nutrients-13-04060-f006:**
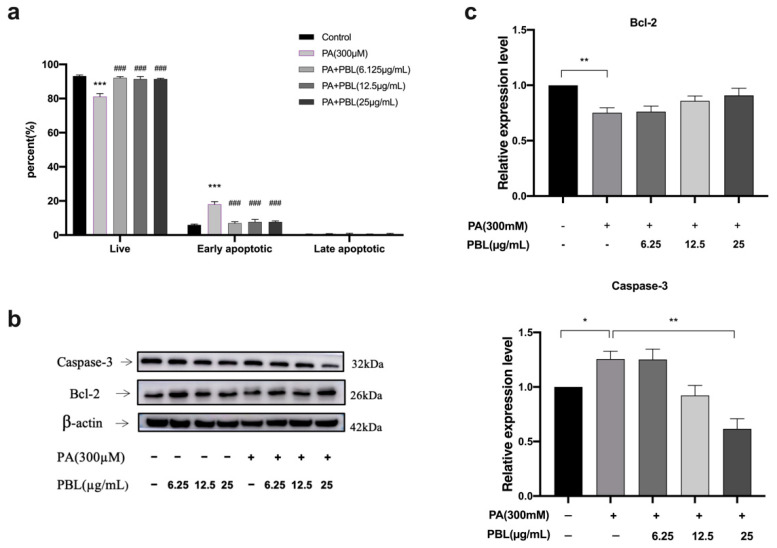
PBL inhibits apoptosis in NAFLD-modeled cells. (**a**) Cells were analyzed by flow cytometry, the percentage of apoptotic cells. The suppression on apoptosis of PBL on NAFLD-modeled cells. The NAFLD-modeled cells were treated with PBL (6.25–25 µg/mL) for 48 h. Data are presented as mean ± SEM (*n* = 3). *** *p* < 0.001 compared with the control group. **^###^**
*p* < 0.01 compared with the PA (300 µM) group. (**b**,**c**) Effects of PBL on Bcl-2, caspase-3 and protein expression in NAFLD-modeled cells. The data are represented as mean ± SEM (*n* = 3). * *p* < 0.05, ** *p* < 0.01 compared between the marked groups.

## Data Availability

Not applicable.

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
