# Peer review of "Network Pharmacology Exploration Reveals Anti-Apoptosis as a Common Therapeutic Mechanism for Non-Alcoholic Fatty Liver Disease Treated with Blueberry Leaf Polyphenols"

_nutrients, 2021, doi:10.3390/nu13114060_

Round 1
Reviewer 1 Report
Line 48-52: Blueberry fruit and blueberry leaves are two distinct entities in terms of potential bioactive compounds. The previously published health beneficial properties associated to blueberry fruits cannot be applied to extracts of blueberry leaves. This distinction should be made much more explicitly in the introduction to avoid ambiguities. Moreover, to support the beneficial effects of solvent extract of blueberry leaves, please cite only those papers that are strictly relevant to this product.
In Chapter 2.2 the Authors claim that information on main blueberry bioactive compounds were explored in a previous research. The cited paper [ref 19] however, focuses on blueberry fruits, not blueberry leaves and extraction was made by MeOH not petroleum ether as indicated in an other cited literature [ref 23]. Please clarify what type of sample (what plant part and what type of extraction) was exactly tested in the cell experiments. Also add how the actual suspension was made that was used to treat cells. Also explain what do you mean by the concentration of PBL used for cell treatment.
The authors claim, the PBL preparation they used for cell treatment contains clorogenic acids and flavonoid glycosides e.g. rutin (quercetin-3-O-rutinoside), isoquercitrin, iridin and anthocyanidin-glucosides such as cyanidin-3-O-glucoside. My major concern about the presented experimental setup and derived reasoning is that these compounds are most probably are not exposed to liver cells, since their bioavailability is moderate. Please give solid and convincing evidence that treatment of HepG2 cells with such glycosylated compounds and caffeoyl-quinicacids (chlorogenic acids) can be considered relevant from a physiological point of view. For this reasoning, please provide also concentrations of these compounds in the final solution used for cell treatment. Compare this with concentrations most probably present in a similar situation in vivo after consuming a realistic portion or dose of PBL product.
Minor remarks:
Line 180: D-catechin and L-epicatechin is not considered correct indication of the given two compounds. These two epimers are most probably: (+)-catechin and (-)-epicatechin.
The following link in chapter 2.2. is not valid: https://ibts.hkbu.edu.hk/LSP/tcmsp.php
Figure 1.: This figure is visually attractive but does not add significant novelty to the written parts. In particular, it is really difficult to detect the actual connections in the network in Figure 1B. If this is considered important then the quality of the figure should be improved, or otherwise, if this aspect is not important than please provide the highlights that validate the inclusion of this figure.
Figure 2.: Procyanidin B1 and kaempferol are indicated in this figure however, these compounds are not quantified. This problem is partly addressed in Chapter 1.3. Please clarify and add some extra supporting information and details on this issue.
Author Response
Dear Editor:
Manuscript ID: nutrients - on 1433029 “Network Pharmacology Exploration Reveals Anti-apoptosis as a Common Therapeutic Mechanism for Non-alcoholic Fatty Liver Disease Treated with Blueberry Leaf Polyphenols”
Thank you very much for your advice on our above manuscript. Please kindly see below our response to the reviewers’ comments as well as our revisions in the attached re-submitted manuscript following most of the comments. We hope the manuscript might now be acceptable and look forwarding to receiving your decision.
With kind regards and best wishes,
Yours sincerely
Wei Zhang
Corresponding Author
Response to the reviewer's comments:
- Line 48-52: Blueberry fruit and blueberry leaves are two distinct entities in terms of potential bioactive compounds. The previously published health beneficial properties associated to blueberry fruits cannot be applied to extracts of blueberry leaves. This distinction should be made much more explicitly in the introduction to avoid ambiguities. Moreover, to support the beneficial effects of solvent extract of blueberry leaves, please cite only those papers that are strictly relevant to this product.
Response: Thank you for your comment. More detailed information has already been added accordingly in Lines 51-54.
- In Chapter 2.2 the Authors claim that information on main blueberry bioactive compounds were explored in a previous research. The cited paper [ref 19] however, focuses on blueberry fruits, not blueberry leaves and extraction was made by MeOH not petroleum ether as indicated in an other cited literature [ref 23]. Please clarify what type of sample (what plant part and what type of extraction) was exactly tested in the cell experiments. Also add how the actual suspension was made that was used to treat cells. Also explain what do you mean by the concentration of PBL used for cell treatment.
Response: Thank you for your comment. Firstly, the preparation method of PBL was to pound the blueberry leaves and extract with s petroleum ether at reflux for 30 min. The filtrate was extracted with an equal volume of 70% ethanol at reflux for 90 min and then concentrated and purified by a macroporous resin column. Three eluents of 30%, 50%, and 70% ethanol eluent were collected and lyophilized into powder. In the cell experiments, the PBL sample was dissolved with DMSO to a final concentration about 100 mg/mL. Then, cells were treated with PBL at the concentrations of 6.25, 12.5 and 25 µg/mL in the experiments of Oil Red O staining, flow cytometry and western blot analysis.
- The authors claim, the PBL preparation they used for cell treatment contains clorogenic acids and flavonoid glycosides e.g. rutin (quercetin-3-O-rutinoside), isoquercitrin, iridin and anthocyanidin-glucosides such as cyanidin-3-O-glucoside. My major concern about the presented experimental setup and derived reasoning is that these compounds are most probably are not exposed to liver cells, since their bioavailability is moderate. Please give solid and convincing evidence that treatment of HepG2 cells with such glycosylated compounds and caffeoyl-quinicacids (chlorogenic acids) can be considered relevant from a physiological point of view. For this reasoning, please provide also concentrations of these compounds in the final solution used for cell treatment. Compare this with concentrations most probably present in a similar situation in vivo after consuming a realistic portion or dose of PBL product.
Response: Thank you for your comment. The effective components of blueberry leaves were selected based on the demands of both OB ≥ 30% and DL ≥ 0.18. Although these compounds bioavailability is moderate, some literatures reported that they have diverse biological activities such as anti-tumor, anti-inflammatory, anti-coagulant. For example, isoquercitrin (IQ) could significantly reduce hepatic lipid accumulation and decrease inflammation and oxidative stress on a high-fat diet (HFD) induced NAFLD models (ref.1). Chlorogenic acid ameliorated liver injury and insulin resistance by suppressing autophagy via inactivation of JNK pathway in NAFLD rat model (ref.2). As shown in Table 1, the content of 10 compounds in the PBL had been displayed. According to these data, we deduced that PBL concentration was less than 2 µM in the cell experiment treatment. Finally, the aim of this study is to elucidate the multi-target mechanism of polyphenols from blueberry leaf extract on NAFLD through network pharmacology and to validate its results via biological experiments, we will pay more attention to single compounds being screened in the treatment of NAFLD in further study.
Reference:
- Qin, G.; Ma, J.; Huang, Q.; Yin, H.; Han, J.; Li, M.; Deng, Y.; Wang, B.; Hassan, W.; Shang, J. Isoquercetin Improves Hepatic Lipid Accumulation by Activating AMPK Pathway and Suppressing TGF-β Signaling on an HFD-Induced Nonalcoholic Fatty Liver Disease Rat Model. Int J Mol Sci 2018,
- Yan H, Gao Y Q, Zhang Y, et al. Chlorogenic acid alleviates autophagy and insulin resistance by suppressing JNK pathway in a rat model of nonalcoholic fatty liver disease[J]. Journal of biosciences, 2018, 43(2): 287-294.
- Ferlemi, AV., Mermigki, P.G., Makri, O.E. et al. Cerebral Area Differential Redox Response of Neonatal Rats to Selenite-Induced Oxidative Stress and to Concurrent Administration of Highbush Blueberry Leaf Polyphenols. Neurochem Res 40, 2280–2292 (2015).
Table 1.
Content of 10 compounds in PBL (25µg/mL).
Compounds |
Concentration(nM) in PBL (25µg/mL) |
(+)-Catechin |
210.25 |
Chlorogenic Acid |
2188.5 |
(-)-Epicatechin |
240.25 |
Isoquercitrin |
1025.75 |
Iridin |
42.75 |
Quercetin |
543.25 |
Cyanidin-3-O-glucoside |
251.5 |
Rutin |
718.75 |
Procyanidin B1 |
10.45 |
Kaempferol |
28.65 |
Minor remarks:
1.Line 180: D-catechin and L-epicatechin is not considered correct indication of the given two compounds. These two epimers are most probably: (+)-catechin and (-)-epicatechin.
Response: Thank you for your comment. The D-catechin and L-epicatechin correct indications have been checked and corrected accordingly in Lines 180-181.
2.The following link in chapter 2.2. is not valid: https://ibts.hkbu.edu.hk/LSP/tcmsp.php
Response: Thank you for your comment. We provided a valid link in the Line 90.
3.Figure 1.: This figure is visually attractive but does not add significant novelty to the written parts. In particular, it is really difficult to detect the actual connections in the network in Figure 1B. If this is considered important then the quality of the figure should be improved, or otherwise, if this aspect is not important than please provide the highlights that validate the inclusion of this figure.
Response: Thank you for your comment. We provided more details in the Line 208-209, accordingly.
4.Figure 2.: Procyanidin B1 and kaempferol are indicated in this figure however, these compounds are not quantified. This problem is partly addressed in Chapter 1.3. Please clarify and add some extra supporting information and details on this issue.
Response: Thank you for your comment. We have added the quantified of procyanidin B1 and kaempferol in PBL. PBL were separated on a Sepax GP-C18 column (2.1 mm × 150 mm, 1.8 μm) at a flow rate of 0.25 mL/min using a gradient mobile phase of 0.1% formic acid and acetonitrile. The procyanidin B1 and kaempferol of PBL (10 µg/mL) were 4.18 nM and 11.46 nM, respectively.

Reviewer 2 Report
I have no comments about the manuscript in its current form.
Author Response
Dear editor and reviewer:
Manuscript ID: nutrients - on 1433029 “Network Pharmacology Exploration Reveals Anti-apoptosis as a Common Therapeutic Mechanism for Non-alcoholic Fatty Liver Disease Treated with Blueberry Leaf Polyphenols”
Thank you very much for your time on our manuscript. We deeply appreciate your recognition of our research work. We hope the manuscript might now be acceptable and look forwarding to receiving your decision.
With kind regards and best wishes,
Yours sincerely
Wei Zhang
Corresponding Author

Round 2
Reviewer 1 Report
Some of my concerns were convincingly responded. However, regarding my major concern, namely that investigated compounds are most probably are not exposed to liver cells at high concentrations, is still not responded. I asked to give solid and convincing evidence that treatment of HepG2 cells with low micromolar concentrations of such glycosylated compounds and caffeoyl-quinicacids (chlorogenic acids) can be considered relevant from a physiological point of view. The authors referred to (ref 1 and ref 2) Both papers present animal studies, which are highly relevant. However, as I referred to this in my previous revision, the observed biological effect is most probably cannot be associated to the native (fed) polyphenols, since their bioavailability is moderate and are not present at low micromolar range in systematic circulation. Most probably, their intestinal metabolites or other indirect effects can be associated with the observed biological effects. Therefore, direct treatment of liver cells with the native compounds is not a highly relevant biological experiment to validate the proposed mechanisms of native polyphenols discovered by the means of network pharmacology. I would like to emphasise that I fully accept the elegantly discovered and proposed multi-target mechanisms, however I doubt the relevancy of the experimental part of the study to validate these findings. The authors should explicitly include this weakness of their study in the text.
Author Response
Dear editor and reviewer:
Manuscript ID: nutrients- on 1433029 “Network Pharmacology Exploration Reveals Anti-apoptosis as a Common Therapeutic Mechanism for Non-alcoholic Fatty Liver Disease Treated with Blueberry Leaf Polyphenols”
Thank you very much for your advice on our above manuscript. Please kindly see below our response to the reviewers’ comments as well as our revisions in the attached re-submitted manuscript following most of the comments. We hope the manuscript might now be acceptable and look forwarding to receiving your decision.
With kind regards and best wishes,
Yours sincerely
Wei Zhang
Corresponding Author
Response to the reviewer's comments:
Some of my concerns were convincingly responded. However, regarding my major concern, namely that investigated compounds are most probably are not exposed to liver cells at high concentrations, is still not responded. I asked to give solid and convincing evidence that treatment of HepG2 cells with low micromolar concentrations of such glycosylated compounds and caffeoyl-quinicacids (chlorogenic acids) can be considered relevant from a physiological point of view. The authors referred to (ref 1 and ref 2) Both papers present animal studies, which are highly relevant. However, as I referred to this in my previous revision, the observed biological effect is most probably cannot be associated to the native (fed) polyphenols, since their bioavailability is moderate and are not present at low micromolar range in systematic circulation. Most probably, their intestinal metabolites or other indirect effects can be associated with the observed biological effects. Therefore, direct treatment of liver cells with the native compounds is not a highly relevant biological experiment to validate the proposed mechanisms of native polyphenols discovered by the means of network pharmacology. I would like to emphasise that I fully accept the elegantly discovered and proposed multi-target mechanisms, however I doubt the relevancy of the experimental part of the study to validate these findings. The authors should explicitly include this weakness of their study in the text.
Response: Thank you for your helpful advice. The aim of our study was to preliminarily elucidate the multi-target mechanism of PBL on NAFLD. Base on network pharmacology analysis, it could demonstrate the relationship between key targets and pathways of the main active ingredient in PBL for the treatment of NAFLD and in vitro biological experimental verification. However, there’s still some limitation of in vitro experiments since the bioavailability of some polyphenols is moderate. Thus, we will focus on the relevant in vivo study and provide deeper insight into the hepatoprotective of PBL in further study. We have added the weakness of our study in the revised manuscript. Again, thank you for your advice.
